# County Scale Corn Yield Estimation Based on Multi-Source Data in Liaoning Province

Ge Qu [1], Yanmin Shuai [2,3,4,*], Congying Shao [1], Xiuyuan Peng [5] and Jiapeng Huang [1]

1 College of Surveying and Mapping and Geographic Science, Liaoning Technical University, Fuxin 123000, China
2 College of Geography and Environmental Sciences, Zhejiang Normal University, Jinhua 321004, China
3 College of Resources and Environmental, University of Chinese Academy of Sciences, Beijing 100049, China
4 CAS Research Center for Ecology and Environment of Central Asia, Urumqi 830011, China
5 Information Research Institute, Liaoning Academy of Agricultural Sciences, Shenyang 110161, China
* Correspondence: shuaiym@ms.xjb.ac.cn

**Abstract:** Corn as a dominant and productive cereal crop has been recognized as indispensable to the global food system and industrial raw materials. China's corn consumption reached $2.82 \times 10^8$ t in 2021, but its production was only $2.65 \times 10^8$ t, and China's corn industry is still in short supply. Timely and reliable corn yield estimation at a large scale is imperative and prerequisite to prevent climate risk and meet the growing demand for corn. While crop growth models are well suited to simulate yield formation, they lack the ability to provide fast and accurate estimates of large-scale yields, owing to the sheer quantity of data they require for parameterization. This study was conducted in the typical rain-fed corn belt, Liaoning province, to evaluate the applicability of our modeling practices. We developed the factors using climate data and MCD43A4 production, and built a county-level corn yield estimation model based on correlation analysis and corn growth mechanisms. We used corn yield data from the county between 2007 and 2017, leaving out 2017 for verification. The results show that our model, with an $R^2$ (the Coefficient of Determination) of 0.82 and an *RMSE* (Root Mean Square Error) of 279.33 kg/hm$^2$, significantly improved estimation accuracy compared to only using historical records and climate data. Our model's $R^2$ was 0.34 higher than the trend yield estimation model and 0.27 higher than the climate yield estimation model. Additionally, *RMSE* was reduced by 300–400 kg/hm$^2$ compared to the other two models. The improvement in performance achieved by adding remote sensing information to the model was due to the inclusion of variables such as monitored corn growth state, which corrected the model predictions. Our work demonstrates a simple, scalable, and accurate method for timely estimation of corn yield at the county level with publicly available multiple-source data, which can potentially be employed in situations with sparse ground data for estimating crop yields.

**Keywords:** yield estimation; multi-source data; climate suitability; corn mechanism

## 1. Introduction

Corn is one of China's most important food crops and a significant forage grain. To meet the growing needs of animal husbandry, the corn industry, rations, and exports, corn is becoming increasingly critical in China's grain production [1]. As can be seen from the changes in the proportion of the total planted area under significant crops in China over the past 30 years, the proportion of rice has increased by 13.4%, wheat has increased by 10.3%, and soybeans has decreased by 43.1%, while corn has increased by 43.8% over the same period, indicating that China's corn is developing exceptionally rapidly [2,3]. In 2021, China's corn consumption reached $2.82 \times 10^8$ t, but its production is only $2.65 \times 10^8$ t, and China's corn industry is still in short supply [4,5]. Timely and accurate crop production estimation can provide information support for national food policy formulation and

market price regulation, which is essential for developing the rural economy and foreign food trade.

Most traditional methods of estimating corn yield rely on ground sampling estimates, which are laborious, destructive, and challenging to scale up to large scales. Therefore, more effective approaches to yield estimation are being explored. In recent years, yield estimation methods have been classified into two groups, process-based crop simulation models and statistical models based on regression algorithms [6–8]. Both groups of models have their pros and cons. Process-based models dynamically simulate the biological parameters of crop growth and yield in specific time steps at a single point scale, such as Decision Support System for Agrotechnology Transfer (DSSAT) [9], Agricultural Production Systems sIMulator (APSIM) [10], and WOrld FOod STudies (WOFOST) [11]. However, these models require extensive and detailed input data on soils, field management measures, and current climate conditions, making the results vulnerable to external interference [12]. Additionally, due to the insufficiency of masses of input data on management practices, it is difficult to use crop growth models to monitor crop growth and estimate yield at regional scales [13]. In contrast, statistical models have fewer inputs, and can acquire rapid and effective crop productivity information, which is critical for understanding of yield variability at regional scale farm management and decision making [14–16]. However, few studies have assessed the sensitivity of estimation factors to crop yields during different phenological periods [17,18]. This operation could enhance the phenological significance of model features. One considerable solution is to master features knowledge about yield estimation through exploration.

Corn is a $C_4$ crop that is more sensitive to climate changes (i.e., temperature, rainfall, and sunshine) than other crops [19]. Therefore, it is essential to consider climatic conditions in crop yield estimation [12,20]. In a study conducted by Lobell D B et al. [21], climatic factors, such as monthly temperature and precipitation, were used to evaluate the impact of climate on the production of multiple staple crops during 1980–2008. The results confirmed that the yield loss caused by climate variation has gradually offset the yield increase realized by technological progress. Despite these findings, climate fluctuation could not fully explain yield variation [18,22]. Additionally, the complex and nonlinear effects of climatic factors on crop growth introduce significant uncertainty [22] in yield prediction, which will bring about a large deviation in estimation results.

Combining multi-source data such as climate and remote sensing can improve yield predictions [23,24]. A vegetation index (VI) is a unique spectral signal extracted from the optical parameters of the leaf canopy and contains over 90% of the remote sensing information for vegetation, making it a valuable tool for crop change studies [25]. The Normalized Difference Vegetation Index (NDVI) is the most popular VI, but the sensitivity of the NDVI decreases when vegetation cover is high [26]. The Green Normalized Difference Vegetation Index (GNDVI) is highly correlated with nitrogen and is suitable for characterizing the canopy biomass of crops at the heading and tasseling stages [27]. The Enhanced Vegetation Index (EVI) enhances the vegetation signal by adding a Blue Band, which increases sensitivity to high biomass areas [28]. The Normalized Difference Water Index (NDWI) can respond to water stress in the crop canopy in a timely manner, which is significant for drought monitoring [29]. Although past studies have verified that VIs can describe crop growth and capture yield changes, there are few discussions on which VI is more effective for crop yield estimation.

While many features can be used for yield estimation, only a few have a direct effect [30,31]. The formation of corn yield is a long process from sowing to kernel maturity, and climate factors and vegetation indexes at different periods affect yield variability to varying degrees [32]. Many yield estimation studies commonly use full-fertility indicators [33]. For example, Mateo-Sanchis A. et al. [34] aggregated all observations before the harvesting time and developed three specific models for corn, soybean, and wheat to evaluate in scenarios of crop yield estimation. However, existing yield estimation schemes have done little work on screening the best indicators and reproductive stages based on

crop growth mechanisms, neglecting the response of pre-growth, mid-growth, and post-growth stages to yield. Further exploration of yield estimation using indicators of different phenological stages is needed. In addition, current research on regional yield estimation application models generally focuses on the nation and province (state) scale, with less attention paid to the county level [35]. Therefore, there is a pressing need for an effective yield estimation model applicable to the county level. As a central grain-producing province, Liaoning is mainly rain-fed and sensitive to climate change [36], making it a suitable area to demonstrate our model's development.

The rest of this paper is structured as follows. Section 2 presents an overview of this study area and data. In Section 3, we present the model building process. Section 4 presents the county-level yield estimation model, and the discussion is presented in Section 5. Finally, we conclude this work and propound the proposal for future development.

## 2. Materials and Methods

### 2.1. Study Area

Our study was carried out in Liaoning Province, located in southern Northeast China (38°–43° N, 118°–125° E). The study area, covering $1.48 \times 10^5$ km$^2$, is shown in Figure 1. The topography and landforms in Liaoning Province are complex, with hummock slopes primarily in the east and west, higher altitudes in the north connecting with the Inner Mongolia Plateau, and mostly plain areas in the middle to the south. The province is bordered to the south by the Yellow Sea and the Bohai Sea.

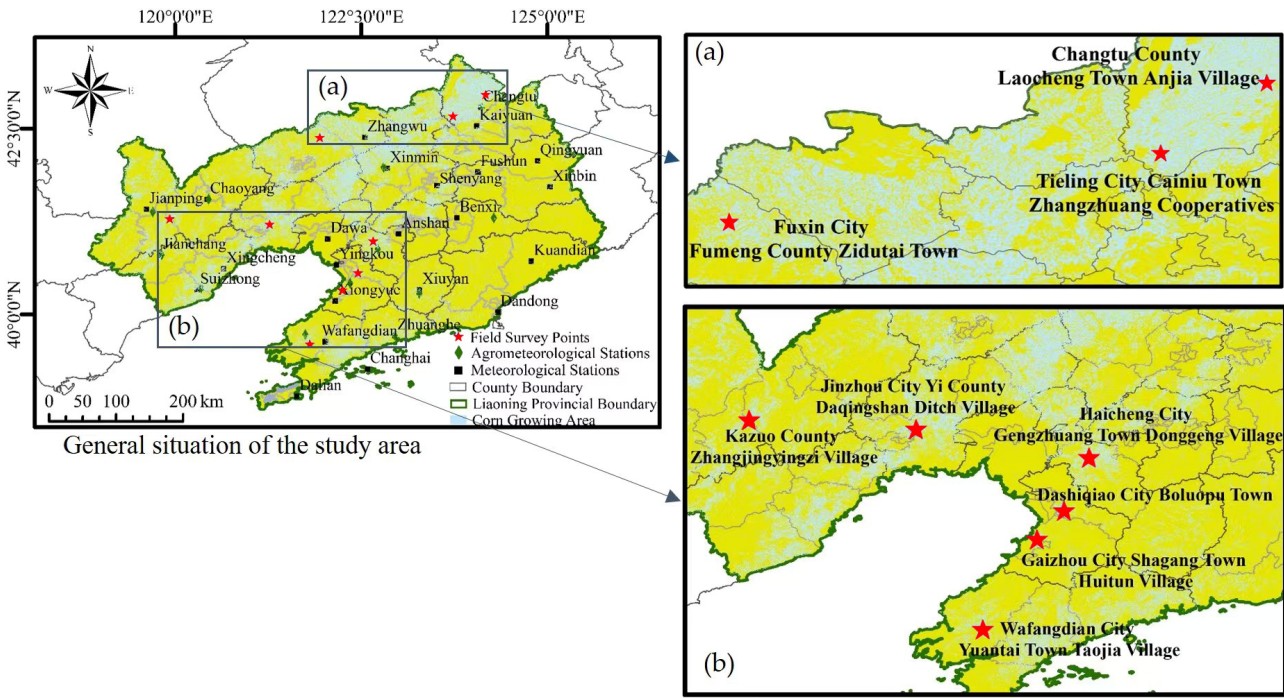

**Figure 1.** Distribution of (**a**,**b**) field survey points, agrometeorological stations, meteorological stations, and corn planting in the study area.

Due to its unique geographical location and complex topography, Liaoning Province has a sensitive and diverse climate. The average annual temperature ranges from 5 to 11 °C, gradually decreasing from the coast to the inland, with a temperature difference of up to 6 °C between north and south. The Liaodong Peninsula and coastal areas have temperatures ranging from 9 to 11 °C, with the highest temperature of 11 °C found in southern Dalian, while Northern Liaoning has temperatures lower than 6 °C near Xifeng County. The average annual precipitation ranges from 439 to 1150 mm and gradually

decreases from southeast to northwest. Counties near Dandong City receive the highest rainfall, reaching about 1000 mm, while counties near Chaoyang City receive the least rainfall, especially in the border area, where it ranges from 450 to 500 mm. The annual sunshine hours range from 2139 to 2938 h sunshine in the west and less in the east. The west bank of Liaodong Peninsula receives more than 2700 h of sunshine, while the west mountain area receives less than 2400 h [36].

Liaoning's rainy season and high-temperature season occur at the same stage, and light resources are abundant. This climate condition is suitable for the development of corn. Furthermore, corn in Liaoning has a long growing season to accumulate sufficient nutrients. The favorable natural conditions have made Liaoning one of country's leading corn areas, with annual corn output increasing from $2.53 \times 10^6$ t to $2 \times 10^7$ t between 1970 and 2021, showcasing its excellent production potential [37]. However, according to climate data of the past 50 years, the temperature in Liaoning has been gradually rising, and the warming–drying trend may have adverse effects on the physiological process of corn.

### 2.2. Data Sources

The remote sensing, climate, corn yield data were obtained from various sources. Details on the data type can be found in Table 1.

**Table 1.** Summary of the collected datasets.

| Category | Name | Spatial Resolution | Temporal Resolution | Source |
|---|---|---|---|---|
| Remote Sensing data | MCD43A4 (Version 6) | 500 m | Daily | Download on the Google Earth Engine (Dataset Provider: NASA LP DAAC at the USGS EROS Center) |
| | Crop Type Cover Products | 10 m | Year | Provided by You et al. [38] (https://doi.org/10.6084/m9.figshare.13090442, accessed on 16 October 2020) |
| Climate data | Precipitation data | Weather Stations | Daily | China Meteorological Administration (http://data.cma.cn, accessed on 20 October 2021) |
| | Temperature data | Weather Stations | Daily | Same as Above |
| | Sunshine data | Weather Stations | Daily | Same as Above |
| Corn Yield data | County-level Corn Yield in Liaoning | County | Year | County Statistical Yearbook |
| | Measured Yield | Field Scale | Year | Provided by Liaoning Academy of Agricultural Sciences |

### 2.2.1. Corn Yield Data

County-level yield data (kg/hm$^2$) refers to the average unit area yield of each county from 2007 to 2017 and were collected from the local County Statistical Yearbook. The data for 2007–2016 were used to build the model, while the data for 2017 were used to validate the model's performance. Acquired through the Liaoning Provincial Bureau of Statistics, the yield data were obtained through a scientific method that combines external harvesting, weighing of sample surveys, and internal analytical operations. These data are among the few yield data available that have area attributes. However, due to confusion in county-level agricultural census pre-record, retrospective validation of fundamental data from the previous 10 years was conducted by the State Administration of Grain and the Chinese Academy of Agricultural Sciences in 2017.

The Liaoning Academy of Agricultural Sciences provided yield records from field measurements to investigate the responses of vegetation index to yield gradients. The measurement sites included Kazuo County Zhangjingyingzi Village (divided into flat and sloping cultivation sample areas), Wafangdian City Yuantai Town Taojia Village, Haicheng City Gengzhuang Town Donggeng Village, and so on (locations shown in Figure 1). During the harvest period of each experimental sample area, corn was reaped using a sickle, threshed, dried, weighed, and converted into yield (kg/hm$^2$) of each plot according to planting density.

### 2.2.2. Climate Data

We used continuous daily climate data to assess the impact of climate resources on corn growth in Liaoning Province. The climate data were obtained from the National Meteorological Center (NMC) for climate suitability calculation, collected from 25 weather stations throughout the province (Figure 1). Among them, the daily maximum, daily minimum, and daily average temperature elements related to temperature, the daily total precipitation related to precipitation, the sunshine hours related to light, the daily average wind speed at 2 m related to wind speed, and so on, were selected [39].

Pre-processing of climate data is required prior to analysis and evaluation, which is essential for subsequent data analysis and modeling. Most of the raw data usually contain some noise, such as: missing values, discrete values (outliers), etc. First, we removed the noise from the climate data according to the data description file. Then, to address missing temperature data resulting from instrumentation problems, the average values from the previous and next day were used to replace the missing values. The missing multi-day data was linearly interpolated using the same period of the nearest station. However, the missing precipitation, sunshine, humidity, and wind speed data were interpolated with the estimated values of the same period in adjacent years based on local seasonal characteristics [40,41].

### 2.2.3. Remote Sensing Data

This study utilized reflectance data from the MODIS "land" bands 1–7 of the MCD43A4 V6 Nadir Bidirectional Reflectance Distribution Function Adjusted Reflectance (NBAR) product [42]. The spatial resolution was 500 m, and the temporal resolution was daily. To obtain cloud-free images of the study area, we de-clouded the multi-temporal images from 2007 to 2017 in the Google Earth Engine (GEE) platform. The MCD43A4 product's "BRDF_Albedo_Band_Mandatory_Quality" band stored image quality information, which allowed us to obtain reliable pixels by using quality bands [43,44]. Masked pixels are not involved in subsequent calculations to ensure research accuracy. Then the study area boundary data was imported into GEE, and the image results covering the study area were obtained using the clipping function.

Moreover, we used the 10 m resolution crop type cover products of You et al. [38] in Northeast China as an interpretation basis and adjusted previous year data using Landsat RGB images, resulting in the 2007–2017 annual general corn distribution results. To more accurately characterize the changes in surface reflectance of corn during the whole fertility period at the MODIS scale, taking into account the balance between rural crop fragmentation and image quality, we first selected the MODIS pixels covering over 60% of the distribution area of corn [45]. Pixels that did not meet the 60% threshold were screened again and were re-evaluated through visual interpretation to ensure that all planting areas were extracted as accurately as possible.

### 2.3. Methods

In this study, the trend yield model includes county-specific fixed effects, including productivity basis and input in agriculture, etc. Climate suitability indicates the impact of short-term climatic fluctuations on final yield. A vegetation index, obtained by remote sensing inversion, provides a comprehensive reflection of crop growth under various external environments, such as pests, diseases, and human intervention. Effective factors are screened using correlation analysis combined with corn growth characteristics. To further quantify the influences of estimation factors, we calculated the factor weights to obtain the different contributions of multiple variables, when they jointly explain corn yield in Liaoning. Using multiple regression analyses, the multi-source factors with weightings can be incorporated into a county yield estimation model aimed at corn. The main idea of the paper is shown in Figure 2.

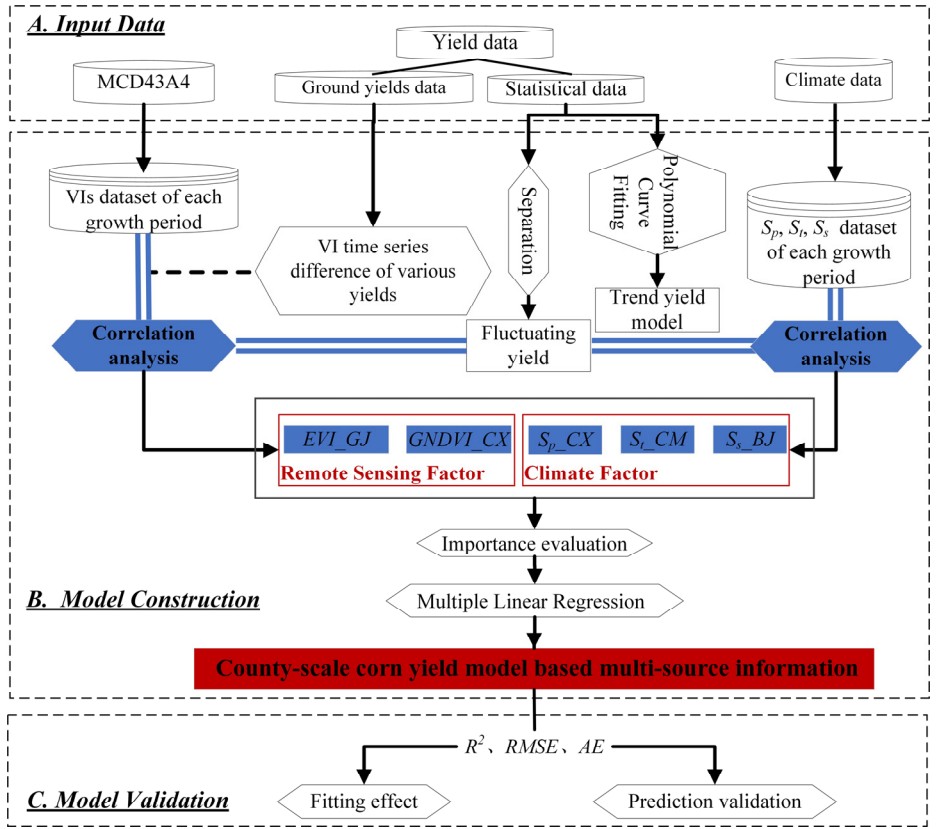

**Figure 2.** Flowchart indicating all steps of the model construction.

2.3.1. Factor Construction

Trend Yield

Crop yield is mainly affected by both long-term fixed effects and short-term fluctuation factors. The yield change caused by social factors and longer-term climate change is commonly referred to as trend yield, which mainly includes long-term technological advancements and social investments. On the other hand, yield variation caused by growing environment is called fluctuation yield, primarily due to short-term weather conditions and plant disease. The variation caused by some other factors is called random yield. Consequently, the total corn yield can be broken down into three distinct components, with the following equation [46]:

$$Y = y_t + y_f + \Delta Y \tag{1}$$

where $Y$ denotes total unit yield value (kg/hm$^2$), $y_t$ denotes trend unit yield (kg/hm$^2$), $y_f$ denotes fluctuating unit yield (kg/hm$^2$), and $\Delta Y$ denotes stochastic yield. Due to the limited perception that the mathematical function cannot express the random yield, and the actual impact is small, it is generally ignored.

Trend model building first required the separation of trend yields based on historical published data (Equations (7) and (8)) [46,47]. The next step was to develop a regression model of trend yields and time (Equation (9)), which allowed the model to predict trends [48,49]:

$$y_i(t) = a_i + b_i \times t \tag{2}$$

$$\overline{y_i}(t) = \frac{1}{q} \times \sum_{j=i}^{q} y_i(t) \tag{3}$$

where $\overline{y_i}(t)$ is the trend yield (kg/hm$^2$) in $t$ year; $y_i(t)$ is the yield (kg/hm$^2$) calculated by fitting a straight line in $t$ year; $a_i$ and $b_i$ are the constants and coefficients of the straight line fitting equation, respectively; $I = n - k + 1$; $k$ is the sliding window size, and $k$ here is 5; $n$ is the number of years of modeled historical data; $t$ is the time series in years; $j = 1, 2, \ldots, q$; $q$ is the number of straight line equations involved in the calculation in $t$ year, taking the value of $1, 2, 3, \ldots k, \ldots, 3, 2, 1$.

$$y_{te} = a_0 + a_1 \times t + a_2 \times t^2 + a_3 \times t^3 \tag{4}$$

where $y_{te}$ is the trend yield (kg/hm$^2$) in the year to be estimated, $t$ is the year to be estimated, $a_1$, $a_2$, and $a_3$ are the polynomial coefficient term and constant term, drawing from the regression fitting.

Climate Factor

Natural factors that constrain corn yields, such as varietal goodness and soil fertility, are long-term, stable factors. Liaoning's corn relies mainly on rain-fed agriculture, and climate fluctuations have a significant impact on crop yields [50]. Therefore, we built the suitability functions on the growth period scale, including precipitation suitability ($S_p$), temperature suitability ($S_t$), and sunshine suitability ($S_s$), in order to reflect the mechanism that climate conditions respond to corn yield change.

The Inverse Distance Weighting (IDW) method was applied to interpolate climate suitability data, calculated using climate data from weather stations [51], in order to achieve the exact spatial resolution of MCD43A4 (500 m). The cross-validation method was used to verify the effect of interpolation. We assumed that the suitability values of each station were unknown, and then interpolated this station using the values of surrounding stations [52]. The $R^2$ between the actual observed and estimated values was 0.83, indicating that the interpolation results are reliable.

In this paper, the parameters in Table 2 were identified based on baseline information on the healthy growth of corn at different fertility stages, mainly including data from China Agrometeorology [53], Food Security Meteorological Services [54], and related literature [41,50,55].

**Table 2.** Parameter values of climate suitability construction at each development stage.

| Fertility Stage | $T_l$ | $T_o$ | $T_h$ | $K_c$ | $b$ |
|---|---|---|---|---|---|
| Sowing Stage | 10 | 17 | 25 | 0.354 | 4.77 |
| Seedling Stage | 14 | 26 | 30 | 0.773 | 5.08 |
| Jointing Stage | 17 | 26 | 33 | 0.773 | 5.08 |
| Tasseling Stage | 16 | 23 | 35 | 1.288 | 5.16 |
| Filling Stage | 14 | 22 | 32 | 1.167 | 5.21 |
| Maturing Stage | 13 | 20 | 30 | 0.615 | 5.24 |

Temperature suitability ($S_t$): Temperature is the critical determinant of crop velocity of progression. Corn is a temperature-loving crop, and exceeding or falling below the growth temperature threshold will affect its developmental rate and impair productivity. The quantitative evaluation system for temperature stress was established through the suitability function [41,50]:

$$S_t = \begin{cases} 0 & T < T_l \\ \frac{(T - T_l) \times (T_h - T)^D}{(T_o - T_l) \times (T_h - T_o)^D} & T_l \leq T \leq T_h \\ 0 & T > T_h \end{cases} \tag{5}$$

where $T$ is the actual temperature (°C); $T_l$ is the minimum temperature threshold for corn growth (°C); $T_h$ is the maximum temperature threshold for corn growth (°C); and $T_o$ is

optimum growth temperature (°C). $D = (T_h - T_o)/(T_o - T_l)$, is a constant for the three base point temperature calculations. The values are shown in Table 2.

Sunshine suitability ($S_s$): Solar radiation is a source of photosynthetic energy for crops, as evidenced by the number of hours of sunlight and intensity. For photophilic crops such as corn, sunshine hours are inclined to delay or postpone the fertility period and better reveal the level of solar radiation participation in the crop growth process. Therefore, from the farming goal of high and steady yield, sunshine hours were regarded as a measure of sunshine suitability for corn. Proceeding from the latitudinal position of Liaoning, the sunshine suitability function, with a relative sunshine duration of 70% as the boundary point, is as follows [41,50]:

$$S_S = \begin{cases} e^{-[(s-s_0)/b]^2} & s < s_0 \\ 1 & s \geq s_0 \end{cases} \tag{6}$$

where, $s$ is the actual sunshine hours (h), $s_0$ is the sunshine hours (h) with a sunshine percentage of 70%, and $b$ is a constant that varies with fertility stage. The values are shown in Table 2.

Precipitation suitability ($S_p$): Liaoning is dominated by rain-fed agriculture. To some extent, rainfall can determine yield harvest or deficit [41,50]. The influence of precipitation is a continuous and dynamic change, and it is more appropriate to bring in the concept of suitability to describe it. Firstly, the precipitation requirements of corn at different stages need to be established and calculated according to the Penman–Monteith formula (P-M) provided by the Food and Agriculture Organization of the United Nations (FAO):

$$ET_c = K_c \cdot ET_0 \tag{7}$$

$$ET_0 = \frac{0.408\Delta\,(R_n - G) + \gamma \dfrac{900}{T+273} U_2 (e_s - e_a)}{\Delta + \gamma(1 + 0.34 U_2)} \tag{8}$$

where $ET_c$ is the crop precipitation requirement (mm·day$^{-1}$), and $K_c$ is the crop coefficient; the values are shown in Table 2. $ET_0$ is the reference transpiration hair (mm·day$^{-1}$), $R_n$ is the net radiation at the crop surface (MJ·m$^{-2}$·day$^{-1}$), $G$ is the soil heat flux (MJ·m$^{-2}$·day$^{-1}$), $T$ is the air temperature (°C), $U_2$ is the wind speed at 2 m height (m·s$^{-1}$), $e_s$ is the saturated water vapor pressure (kPa), $e_a$ is the actual water vapor pressure (kPa), $e_s - e_a$ is the saturated water vapor pressure difference (kPa), $\Delta$ is the slope of the water vapor pressure curve (kPa·°C$^{-1}$), and $\gamma$ is the hygrometer constant (kPa·°C$^{-1}$).

$$S_p = \left\{ \begin{array}{ll} \frac{P}{ET_c} & P < 0.68 ET_c \\ 1 & P \geq 0.68 ET_c \end{array} \right\} \tag{9}$$

$S_p$ is the precipitation suitability, and $P$ is the total precipitation (mm). Since corn in Liaoning is mostly grown in brown soil areas, where precipitation can infiltrate into the soil and be absorbed and stored more quickly, the total precipitation during the fertility period of corn is defined as light drought when the total precipitation is 0.68 times the water requirement and above [41].

Remote Sensing Factor

A vegetation index effectively combines multi-band reflectance information to highlight crop spectral characteristics more prominently. In this study, we calculated 12 vegetation indexes (VI) from 2007 to 2017, which are recognized as demonstrating crop physiological traits such as chlorophyll content and canopy water content. The equations for these indexes are shown in Table 3.

**Table 3.** Vegetation Indexes (For MCD43A4).

| Vegetation Index | Equation |
| --- | --- |
| Normalized Differenced Vegetation Index, NDVI [56] | $\frac{B2-B1}{B2+B1}$ |
| The Enhanced Vegetation Index, EVI [57] | $2.5 \times \frac{B2-B1}{B2+6 \times B1-7.5 \times B3+1}$ |
| Modified Soil Adjusted Vegetation Index, MSAVI [58] | $\frac{\left(2 \times B2+1-\sqrt{(2 \times B2+1)^2-8 \times (B2-B1)}\right)}{2}$ |
| Differenced Vegetation Index, DVI [59] | $B2 - B1$ |
| Green Normalized Differenced Vegetation Index, GNDVI [60] | $\frac{B2-B4}{B2+B4}$ |
| Renormalized Differenced Vegetation Index, RDVI [61] | $(B2-B1)/\sqrt{B2+B1}$ |
| Land Surface Water Index, LSWI [62] | $\frac{B2-B6}{B2+B6}$ |
| Normalized Difference Senescent Vegetation Index, NDSVI [63] | $\frac{B6-B1}{B6+B1}$ |
| Normalized Difference Tillage Index, NDTI [64] | $\frac{B6-B7}{B6+B7}$ |
| Normalized Difference Greenness Index, NDGI [65] | $\frac{B4-B1}{B4+B1}$ |
| Normalized Difference Water Index, NDWI [66] | $\frac{B2-B5}{B2+B5}$ |
| Mid-IR Bispectral Index, MIRBI [67] | $10 \times B7 - 9.8 \times B6 + 2$ |

2.3.2. Climate Factor Selection Based on Correlation

The Correlation Coefficient ($r$) between climate suitability and fluctuating yield ($y_f$ in Equation (1)) can characterize yield response degrees to light, temperature, and water resource utilization (shown in Figure 3). The input climate factors were determined through the correlation analysis combining corn cultivation experience and physiological basis.

The $r$ between $S_p$ at the tasseling stage and fluctuating yield is the highest (Figure 3a), which is mainly caused by a reduction in grain number per ear due to water deficiency, ultimately affecting the final yield. Insufficient water intake during this period can lead to early plant senescence, a sharp decrease in leaf area, and a decrease in the photosynthetic rate of the leaves, even lower than the respiration rate. A water deficit can also result in delayed or absent male spike emergence, leading to tasseling stage delays that may not synchronize with female spike development [68]. Furthermore, the reduction in the number of branches and spikelets due to water deficiency reduces the amount of pollen, resulting in a significant increase in the amount of sterile male spike pollen. The female spike is also affected, making it difficult to extract the filaments, which hinders fertilization and results in shrinkage or even no formation of grains leading to a severe reduction in the number of grains per ear. During the tasseling stage, each day of drought can reduce the yield by approximately 4%, eventually only leaving about 50% of the normal grain yield, and the fruit quality deteriorates steeply [69].

During the seedling stage, the $r$ between $S_t$ and temperature is at its highest, which indicates that this growth period is the most sensitive to temperature changes and has the highest yield variability (as shown in Figure 3b). This correlation can be attributed to the fact that low temperatures decrease the survival rate of corn seeds, which in turn affects the activity of important growth elements. In the Northeast region of China, spring temperatures are often low, leading to cold injury in corn seedlings and delayed emergence. This lag in growth can lead to inadequate balance of water and sunlight, which are crucial for the growth of corn, and make the crop more susceptible to autumn frosts in later growth stages. Low temperatures can also induce crop diseases. When the mean daily temperature is <10 °C, a series of physiological activities and biochemical reactions within the seedling are delayed or disrupted [70]. In addition, leakage of trace elements, such as calcium and potassium can lead to metabolic imbalances in the body and exacerbate damage to the cell

membrane system. This increased membrane permeability allows endosperm nutrients to leak, which provides nutrients for pathogens adapted to low temperatures. Furthermore, low temperature can also hinder protein synthesis by affecting zymogen activation, which in turn affects the differentiation of radicles and plumules [71,72]. The longer the duration of low temperature stress, the higher the rate of seedling wilting or mortality.

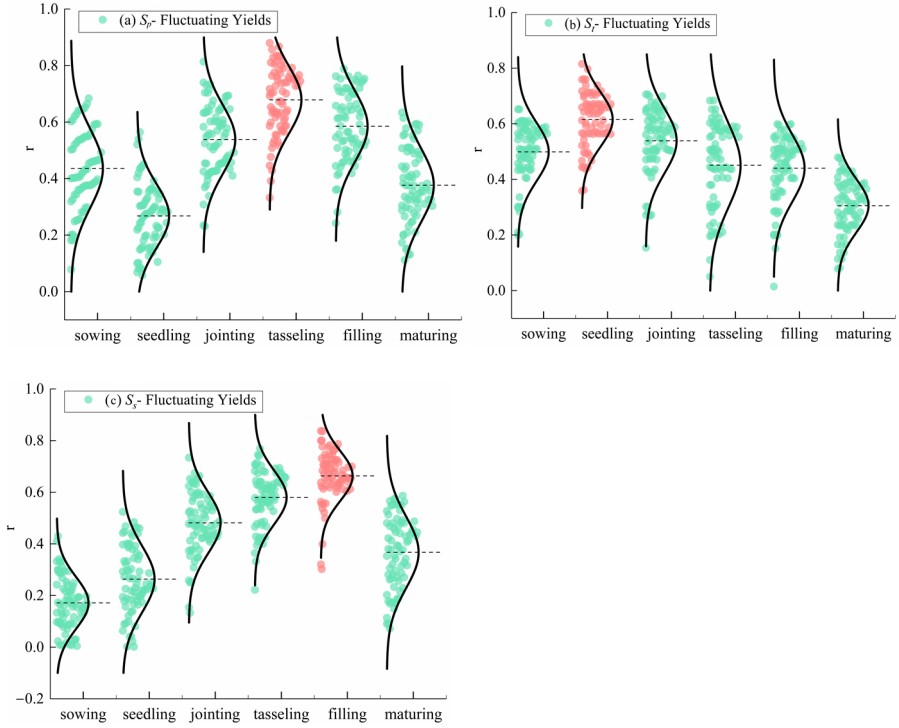

**Figure 3.** Normal distribution comparison of correlation coefficient between (**a**) precipitation suitability, (**b**) Temperature suitability, (**c**) sunshine suitability and fluctuating yield.

The $r$ between $S_s$ at the grain filling stage, is the growth period when sunshine has the greatest influence on the final yield and the fluctuating yield is the highest (Figure 3c). The reduction in yield is mainly caused by a decrease in 100-grain weight due to insufficient light. August is the critical period for dry matter accumulation in corn grain filling, which is a critical period for the formation of economic yield or seed weight. During this period, insufficient light decreases the photosynthetic rate and restricts the synthesis, transport, and storage of starch. This situation results in a reduction in endosperm and grain fullness, ultimately leading to a decrease in the hundred-grain weight. Continuous cloudy conditions will cause the bald area of corn cob to broaden and even increase the probability of no cob cornstalk rise [73].

Based on the above correlation analyses between climatic suitability and fluctuating yield, precipitation suitability at the tasseling stage ($S_p\_CX$), temperature suitability at the seedling stage ($S_t\_CM$), and sunshine suitability at the grain filling stage ($S_s\_GJ$), which are the most critical climatic conditions to return climatic constraints on corn production as realistically as possible, were selected as part of the model yield estimation factors.

### 2.3.3. Remote Sensing Factor Selection Based on Correlation

To form an a priori understanding of spectral response to yield changes, we compared the VI curves of the sample units with yield gradients difference. The VI time series curves of sample units are shown in Figure 4. We found that valid recognizing time, except for MIBRI, occurred in the middle or late growth period which is in accord with the corn weathering pattern. The recognition time of different VIs has different validity. The high-yielding sample units generally have high VI, especially in EVI (Figure 4h), GNDVI

(Figure 4d), and MSAVI (Figure 4k), as they respond more sensitively to yield than water indexes, such as LSWI (Figure 4i) and NDWI (Figure 4j). The VI of sample units indicates that many VIs are related to yield but only a few of them have a direct correlation. Therefore, we expand the research scale to the county and further explore the correlation between the average of the VI for each stage at county scale and fluctuating yield based on corn growth characteristics and determined the optimum VIs for corn yield estimation.

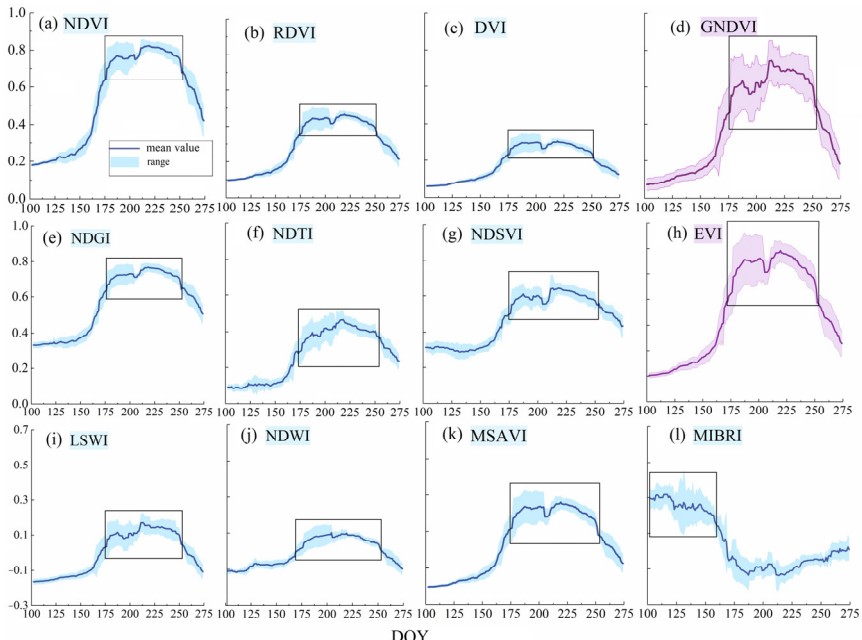

**Figure 4.** Time-series characteristics of vegetation indices in sample areas with different yields (the solid line is the mean, and the shaded area is the range).

The *r* between VI and fluctuating yield is at its minimum from the sowing to the jointing stage (show in Figure 5), with most values ranging from 0.2 to 0.3. During the early stages of corn growth, the plants have a short morphology and small leaves. As a result, VIs, which measure vegetation greenness, cannot effectively monitor the plants. The *r* between water index and fluctuating yield is also low, as corn has weak leaf transpiration in early stages, consuming less water. Even modest drought can help improve late drought resistance, but this leads to an obscure response of LSWI and NDWI to fluctuating yield during the early growing period of corn.

The *r* between VI and fluctuating yield upraise distinctly during the jointing stage, but it is not steady. As corn grows rapidly after jointing, the leaves elongate and widen, and the population leaf area begins to grow linearly, gradually approaching its highest value. However, after reaching a threshold, the leaves begin to shade each other, decreasing the utility rate of luminous energy, and then affecting organic matter accumulation. Therefore, using the VI, which mainly reflects the greenness of corn, to indicate the yield can lead to errors.

During the tasseling, the *r* between VI and fluctuating yield considerably increase compared to the previous period and shows a significant positive correlation. This is particularly evident for GNDVI, as shown in Figure 5b, with values concentrated in the 0.6–0.8 range, indicating a high correlation. The main reason for this increase is that the tasseling stage is a critical period for corn growth, as it marks the transition from nutritional growth to reproductive growth and is also when kernel formation occurs, one of the "three critical corn yield factors". Photosynthesis during this period produces nutrients mainly employed for pollination and insemination. Nutrient deficiencies can lead to poor pollination, as the style will not be able to draw out pollen, resulting in poorly fertilized flowers and significantly reducing the number of normally developed grains. It

is noteworthy that leaves are the central organ of photosynthesis, and leaf area reaches its peak and stability in tasseling. GNDVI replaces the red band preferred by traditional vegetation indexes, such as NDVI, with the green band for a better assessment of crop chlorophyll content [69]. Therefore, GNDVI at the tasseling stage has a significantly positive correlation with the fluctuating yield, indicating that it can effectively monitor corn growth during this critical period.

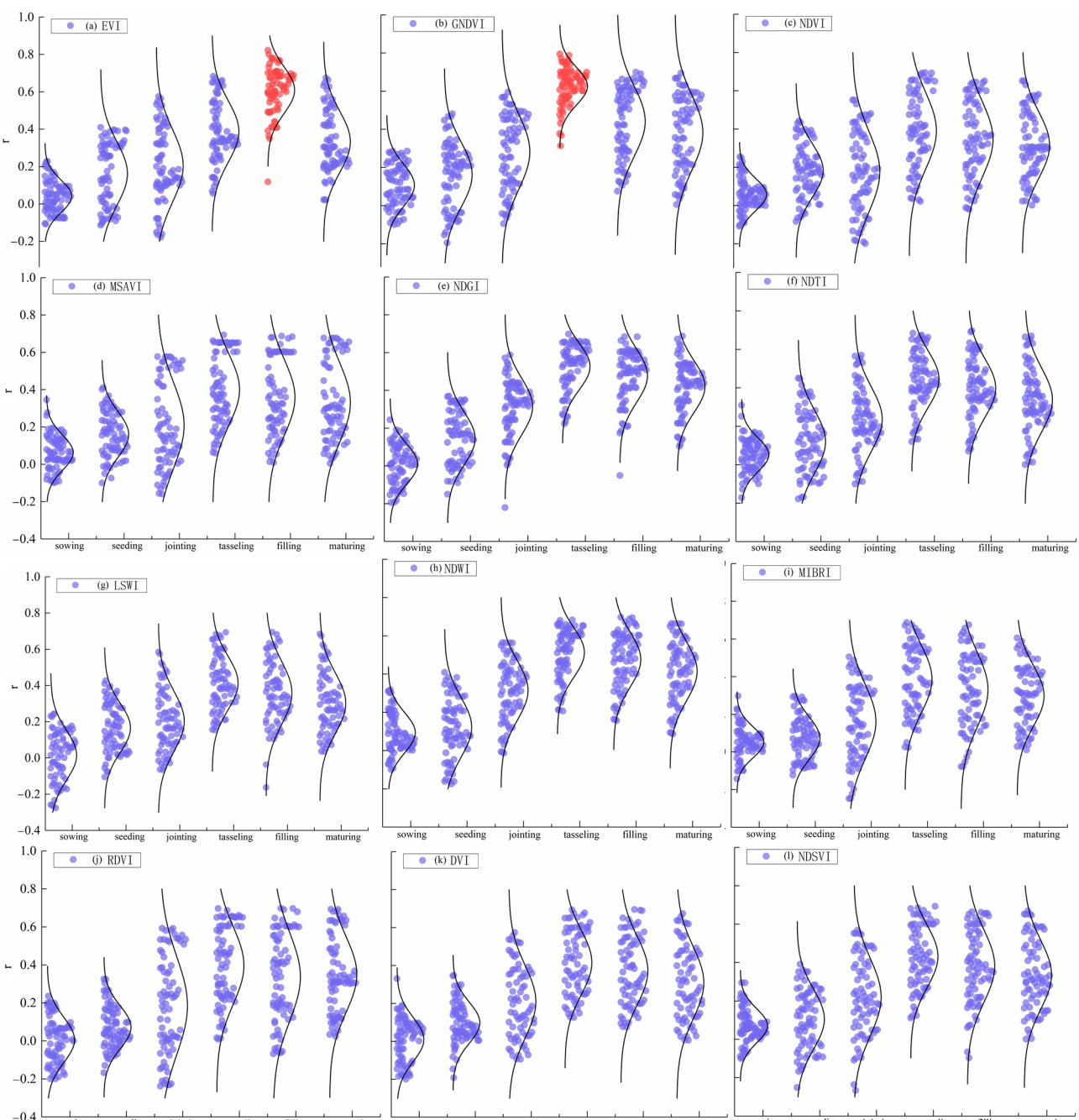

**Figure 5.** Normal distribution comparison of correlation coefficient between different vegetation indices and fluctuating yield.

The *r* between VI and fluctuating yield at the filling stage is second only to the tasseling stage and is also significantly correlated. Among the VI, EVI was more prominent, as shown in Figure 5a. Dry matter accumulation contributes greatly to yield during grain filling, and the duration and rate of grain filling determine the final kernel weight and corn yield.



While genes play a crucial role in regulating grain filling, some observable features have been found to be associated with it, such as a high maximum leaf area and a fast leaf senescence rate. [74]. EVI can better reflect changes in canopy structure, including leaf area index (LAI), and has a deeper light penetration and is not easily saturated in areas with a high leaf area base [75]. Therefore, EVI at the grain filling stage is suitable for describing the grain filling ability and has a high correlation with fluctuating yield.

The *r* between VI and fluctuating yield recedes during the maturing stage, with an *r* value mainly between 0.2 and 0.4. This is likely due to the sharp decrease in green leaves of corn plants during this stage. If the green leaves wither and turn yellow too early, it can be detrimental to the accumulation of organic matter in the grain. On the other hand, if the green leaves delay senescence, it will facilitate the continuous delivery of nutrients to the grain, so holding the green state of corn during the maturing period is also one of the means to judge whether a high yield will be reached.

As mentioned earlier, the correlation between the vegetation index and fluctuating yield during the tasseling and filling stages was more remarkable than others. Taking into account the growth characteristics of corn, GNDVI at the tasseling stage (*GNDVI_CX*) and EVI at the filling stage (*EVI_GJ*) were selected to structure the model due to their high correlation coefficients. This provides the immediate growth dynamics of corn under comprehensive environments.

### 2.3.4. Factor Weight Determination for Overall Situation

The variable importance measure of the Random Forest model illustrated the contribution degree of selected factors on the estimation of corn yield for Liaoning. Before fitting the coefficients for each county, we used the fluctuating yields and factors of all counties from 2007 to 2016 as the dataset and obtained the factor weight assignments through the RF model [12]. The localization of yield estimation model in Liaoning was elementarily realized, and then the model was constructed against each county by using the methods of regression. This logical pattern that adjusted the model from overall situation to local is conducive to improving its adaptation. The key evaluation indicators of the Random Forest model were computed: the Explained Variance (*EV*) score was 0.89, the Mean Absolute Error (*MAE*) was 0.45, the Mean Squared Error (*MSE*) was 0.87, and the Coefficient of Determination ($R^2$) was 0.88, suggesting that the model works well, and the importance evaluation results are reliable [13,76].

The $S_p\_CX$ importance score of 30.27 was higher than the other yield estimation factors, thereby supporting the notion that drought poses the most significant challenge to corn production in Liaoning. The next *EVI_GJ* was 25.79, which similarly confirms that grain filling directly affects the final yield, while the remaining $S_t\_CM$, *GNDVI_CX* and $S_s\_BJ$ were 23.48, 21.19 and 20.45, respectively. Normalized according to the importance scores, the weighting coefficients of $S_p\_CX$, *EVI_GJ*, $S_t\_CM$, *GNDVI_CX* and $S_s\_BJ$, were 0.25, 0.21, 0.19, 0.18 and 0.17, respectively.

## 3. Result

### 3.1. County-Level Yield Estimation Model Results

Multiple linear regression is a method used to establish a linear regression equation between the dependent variable and each independent variable. The established equation is then used in prediction [77]. The multiple linear regression model is simple and flexible in prediction, easy to understand and implement. Therefore, it can meet the needs of this study, which aims to build a wide and fast yield estimation model application [78].

After determining the weight of factors targeting overall Liaoning, we use Multiple Linear Regression for each country to establish the linear relationship between the explanatory variables (weighted climate factors and remote sensing factors for a county) and response variables (corresponding fluctuating yield for each country), adding trend yield to complete the final predicting process in building the county-level yield estimation

model [79]. The equations of the estimation model coupled with multiple source data are as follows, and the model coefficients are shown in Table 4.

$$Y_i = y_{te} + y_i(S_pCX, S_tCM, SsBJ, EVIGJ, GNDVICX) \tag{10}$$

$$
\begin{aligned}
y_i(S_pCX, S_tCM, SsBJ, EVIGJ, GNDVICX) = a \times (0.25 \times S_rCX) + \\
b \times (0.21 \times EVIGJ) + c \times (0.19 \times S_tCM) + d \times (0.18 \times GNDVICX) + \\
e \times (0.17 \times S_sBJ) + k
\end{aligned}
\tag{11}
$$

where $Y_i$ is the estimated total yield (kg/hm$^2$) of $i$ county, $y_{te}$ is the estimated trend yield (kg/hm$^2$), $y_i(S_pCX, S_tCM, SsBJ, EVIGJ, GNDVICX)$ is the fluctuating yield (kg/hm$^2$), $a$, $b$, $c$, $d$, and $e$ are the multiple regression coefficient terms, and $k$ is a constant term.

**Table 4.** Parameters of corn coupling yield estimation model in counties.

| Region | Coefficient | a | b | c | d | e | k | Region | Coefficient | a | b | c | d | e | k |
|---|---|---|---|---|---|---|---|---|---|---|---|---|---|---|---|
| Eastern Liaoning | Shuncheng | 195.92 | −273.42 | 91.08 | 250.44 | 149.19 | 199.35 | Western Liaoning | Taihe | −312.18 | 189.83 | −491.35 | 213.84 | 107.79 | 162.99 |
| | Fushun | −222.51 | −307.47 | 32.15 | 242.09 | 141.24 | 192.99 | | Heishan | −213.74 | 177.76 | −438.9 | 188.75 | 88.75 | 151 |
| | Pingshan | −231.53 | 283.17 | −216.48 | 357.15 | 213.44 | 250.75 | | Yixian | −359.28 | 305.47 | −366.92 | 326.25 | 226.25 | −261.44 |
| | Xihu | 164.47 | 350.68 | −45.4 | 134.69 | 47.83 | 115.26 | | Linghai | 303.36 | 5.86 | −216.48 | −176.29 | 203.36 | 242.69 |
| | Xinbin | −214.01 | −116.98 | 202.77 | 283.36 | 183.66 | 26.93 | | Beizhen | 480.16 | 189.7 | −274.9 | −358.17 | 258.17 | −286.54 |
| | Qingyuan | 206.21 | 263.11 | −38.22 | −309.51 | 216.72 | 50.85 | | Haizhou | −248.37 | 23.97 | 291.27 | 274.42 | 180.11 | −219.53 |
| | Mingshan | −175.99 | 28.18 | −74.53 | 300.79 | 220.07 | 240.58 | | Xinqiu | −157.99 | 130.44 | −104.5 | −253.85 | 151.48 | 201.18 |
| | Nanfen | 197.08 | 264.25 | −8.54 | −251.78 | 97.08 | 157.66 | | Taiping | −203.71 | 136.22 | −98.37 | 138.25 | 63.94 | 109.55 |
| | Benxi | −203.81 | 256.73 | −104.64 | 313.92 | 242.22 | 259.14 | | Qinghemen | 206.13 | 163.69 | −188.32 | 143.29 | 43.64 | −114.59 |
| | Huanren | −176.49 | 173.84 | 52.03 | 164.7 | 65.86 | 132.69 | | Xihe | 203.38 | 446.62 | 170.44 | 331.7 | −208.57 | 241.25 |
| | Zhenan | −183.9 | 363.62 | 8.54 | 20.96 | −257.36 | 340.33 | | Fumeng | 252.56 | 268.22 | 352.01 | 185.06 | 73.14 | 138.51 |
| | Kuandian | 153.65 | 378.28 | −108.56 | −338.13 | 247.33 | 269.86 | | Zhangwu | −382.21 | 114.23 | 401.79 | 250.42 | 144.12 | 195.54 |
| | Donggang | −198.5 | 89.66 | −16.38 | 178.48 | 73.19 | 142.79 | | Shuangta | −352.47 | 4.16 | 66.22 | 193.68 | 252.47 | 281.97 |
| | Fengcheng | 144.88 | 293.46 | 183.95 | 244.88 | −171.12 | 195.9 | | Longcheng | 240.41 | −303.67 | 311.33 | 301.71 | −248.8 | 241.37 |
| | Gongchangling | 189.19 | 467.84 | −124.14 | 296.52 | 200.5 | 237.22 | | Chaoyang | −197.28 | 274.62 | 383.68 | −281.13 | −146.82 | 197.46 |
| | Taizihe | 212.76 | 491.77 | −12.15 | 17.56 | −147.57 | 198.06 | | Jianping | 330.4 | 194.3 | 71.27 | 214.93 | 230.4 | 264.32 |
| | Liaoyang | −250.94 | 169.85 | 137.36 | 70.93 | 69.85 | 135.88 | | Kazuo | −369.33 | 364.09 | −346.69 | −228 | 264.09 | 291.27 |
| | Dengta | −155.25 | 435.85 | 250.8 | 248.79 | 148.36 | 199.03 | | Beipiao | −167.24 | −67.67 | 124.14 | 162.14 | 70.45 | 129.71 |
| Northern Liaoning | Yinzhou | −170.09 | 55.11 | 241.45 | 209.24 | 104.96 | 163.96 | | Lingyuan | −244.73 | 270.43 | −67.99 | 370.43 | 199.27 | −296.35 |
| | Qinghe | −189.26 | 330.01 | 4.34 | 363.89 | 270.16 | 291.11 | | Lianshan | −197.01 | 177.8 | 342.31 | 180.53 | 84.65 | 144.42 |
| | Tieling | −248.62 | 498.2 | 394.73 | 300.77 | 195.72 | 236.58 | | Longgang | −236.11 | 167.77 | 255.83 | 353.84 | 253.84 | −283.07 |
| | Xifeng | −202.42 | 134.9 | 321.75 | 250.01 | 145.95 | 196.04 | | Nanpiao | −241.59 | 494.91 | 473.58 | 283.06 | 37.36 | 109.89 |
| | Changtu | −167.06 | 41.15 | 349 | 342.2 | 238.45 | 273.76 | | Suizhong | −226 | 346.26 | 247.57 | 273.16 | 178.65 | 222.92 |
| | Diaobingshan | −224.05 | 441.28 | 282.14 | 207.83 | 125.69 | 180.55 | | Jianchang | −157.33 | 83.93 | 151.1 | 66.79 | 168.72 | −294.98 |
| | Kaiyuan | 181.35 | 136.19 | 10.27 | −398.64 | 268.15 | 294.52 | | Xingcheng | −337.84 | 289.06 | −183.95 | −213.3 | 103.38 | 162.71 |
| Southern Liaoning | Ganjingzi | −485.56 | 257.32 | −147.08 | 161 | 61.56 | 128.8 | | Panshan | −190.24 | 35.01 | 82.17 | 33.99 | 264.83 | 291.86 |
| | Lvshun | −199.86 | 28.31 | −136.11 | 276.33 | 173.74 | 219 | Central Liaoning | Sujiatun | −215.21 | 478.17 | −228.54 | −254. 32 | 134.27 | 187.42 |
| | Jinzhou | −226.99 | 269.61 | −246.14 | 347.34 | 231.73 | 265.38 | | Hunnan | −244.18 | 444.13 | −77.95 | −140.79 | 32.87 | 106.3 |
| | Changhai | 330.63 | −224.26 | −280.63 | 299.41 | 191.92 | −233.54 | | Shenbei | −201.67 | 144.54 | −311.58 | 229.06 | 113.53 | 170.82 |
| | Wafangdian | −211.69 | 137.77 | 152.87 | 136.82 | 32.41 | −105.93 | | Yuhong | 166.13 | −246.52 | −320.2 | 40.13 | 160.85 | 208.68 |
| | Pulandian | −210.37 | 292.22 | −185.4 | 247.44 | 129.19 | 183.35 | | Liaozhong | 51.8 | 87.89 | −282.44 | 144.08 | 44.08 | 115.26 |
| | Zhuanghe | −479.1 | 210 | −179.1 | 217.1 | 114.43 | 171.54 | | Kangping | −238.63 | 29.38 | −144.64 | 275.48 | 163.95 | 210.78 |
| | Qianshan | −155.17 | 182.14 | −248.88 | 279.72 | 170.12 | 216.1 | | Faku | −196.27 | 272.76 | −90.37 | −352.5 | 245.04 | 276.03 |
| | Taian | −189.22 | 217.09 | −138.59 | 321.61 | 221.61 | 257.29 | | Xinmin | 248.91 | −446.02 | −24.57 | 46.91 | 243.53 | 275.18 |
| | Xiuyan | −214.08 | 7.56 | −281.77 | 231.04 | 187.33 | 229.86 | | | | | | | | |
| | Haicheng | −161.83 | 425.58 | −301.81 | 344.43 | 244.43 | 275.55 | | | | | | | | |
| | Dawa | −167.48 | 264.4 | 188.98 | 396.24 | 261.76 | 289.41 | | | | | | | | |
| | Bayuquan | 191.55 | 436.91 | −326.66 | 107.01 | −213.52 | −245.61 | | | | | | | | |
| | Laobian | −219.48 | 345.9 | −259.35 | 276.88 | 184.84 | 227.88 | | | | | | | | |
| | Gaizhou | 215.44 | 196.15 | −258.14 | 331.4 | −19.64 | −255.71 | | | | | | | | |
| | Dashiqiao | −252.72 | 482.23 | −339.07 | −153.18 | 50.77 | 122.55 | | | | | | | | |

The coefficients of county-level models showed regional differences. Western Liaoning counties have a higher absolute value of $S_p\_CX$ coefficients while slightly $St\_CM$, which match with climate characteristics in study area and manifest the impact degree of factors on corn production in this county. The distribution of rainfall in Liaoning is extremely uneven, in summer the precipitation in the west is significantly more than that in the east and the drought lasts longer. Furthermore, Liaoning is affected by the monsoon intensifying gradually from mountains to plains, except in extreme years, the frost-free period slightly increases from the northwest to the southeast, the cold season is long, and the end of the last frost is late. Corn emergence occurs at the end of spring, and the temperature in western Liaoning rises late, which reduces the survival rate of seedlings.

The *EVI_GJ* and *GNDVI_CX* coefficients had no apparent spatial trends, and the coefficient magnitudes were relatively random, suggesting the function of remote sensing information as a regulating factor. There is relatively abundant in sunshine resources for Liaoning, and sunshine hours in late summer and autumn are nearly sufficient, which

can basically meet the light demand for corn growth, so the *Ss_BJ* coefficient presents a relatively uniform distribution.

### 3.2. County-Level Yield Estimation Model Performance

We used the historical data back substitution confirmation to evaluate the fitting accuracy of the model and the extrapolation forecast test to evaluate the practical application accuracy of the model. The historical data back substitution confirmation employed the Coefficient of Determination ($R^2$), Root Mean Square Error (*RMSE*), and Absolute Error (*AE*) as model evaluation indicators of fitting precision. In multiple regression analysis, $R^2$ reflects the correlation between the fitted model and the actual corn yield data (the statistics yield from 2006 to 2016). A larger $R^2$ means a higher fitting accuracy. The *RMSE* estimates the accuracy of the model by calculating the deviation between the predicted and actual values, and a smaller value means a better model. The smaller *AE* means the predicted value is closer to the true value. The extrapolation forecast test calculated two indicators, $R^2$ and *RMSE*, to evaluate the effectiveness of the practical application of the model, where the statistical yields not involved in the modeling were taken as true values (the statistics yield from 2017) and the yields predicted using the model were taken as predicted values [80]. We compared the three models at the county scale, including the trend yield estimation model (T-M), the yield estimation model constructed by trend yield and climate factors (T&C-M), and the yield estimation model corrected by adding remote sensing information (T&C&RS-M). The calculation formula is as follows:

$$R^2 = 1 - \frac{\sum\limits_{i=1}^{n} (y_i - \hat{y}_i)^2}{\sum\limits_{i=1}^{n} (y_i - \overline{y}_i)^2} \tag{12}$$

$$RMSE = \sqrt{\frac{\sum\limits_{i=1}^{n} (y_i - \hat{y}_i)^2}{n}} \tag{13}$$

where $y_i$ is the estimated yield, $\hat{y}_i$ is the actual yield of county $i$, and $\overline{y}_i$ is the average of the actual yield.

Figures 6 and 7 display the fitting *AE*, $R^2$, and *RMSE* for T-M, T&C-M, and T&C&RS-M in each county from 2007 to 2016. The *AE* of T-M ranged from basically 37 to 62 kg/hm$^2$, the *AE* of T&C-M ranged from 25 to 39 kg/hm$^2$, and the *AE* of T&C&RS-M was down to 13~29 kg/hm$^2$; clearly illustrating that T&C&RS-M had a smaller estimation error than T-M and T&C-M. From a spatial distribution perspective, there were significant geographical differences in the T-M fitting effect. The $R^2$ of T-M closed to 0.9 in most counties of Shenyang City and Jinzhou City, but diminishing to about 0.58 in Jianping County, Kazuo County, and Jianchang County, while the *RMSE* fluctuation range also expanded to 49.5–583.1 kg/hm$^2$, with poor effect. The $R^2$ of T&C-M improved remarkably with the lowest value increasing to 0.658, and the fluctuation range of *RMSE* narrowed to 46.4–375 kg/hm$^2$. The $R^2$ of T&C&RS-M performed steadily, reaching above 0.8 in all counties, while the *RMSE* was less than 150 kg/hm$^2$.

Besides comparing the performance and generalization of the models, we conduct a "leave-one-year-out" experiment to verify the extrapolation potential of the models, which is building models based on the corn yields in ten out of eleven years and then verifying the estimated yield result of the year left out. The model prediction validation is shown in Figure 8. The mean $R^2$ of T&C&RS-M, T&C-M, and T-M on corn yields in Liaoning which were, respectively, 0.82, 0.55, and 0.48, indicated that the predictive effect of T&C&RS-M significantly improved. The *RMSE* of T-M in Liaoning counties varied in the range of 96.5–617.7 kg/hm$^2$, the *RMSE* of T&C-M was 204.9~553.8 kg/hm$^2$, and the *RMSE* of T&C&RS-M was 204.9~553.8 kg/hm$^2$. The results show that the estimate effect of T&C&RS-M is relatively stable. Spatially, the errors are mainly in the west and north. The

difference of the three models is concentrated in Chaoyang and Huludao cities, where T-M and T&C-M generally overestimate the corn yield in low-yield areas. It can be seen that adding remote sensing information to the model brought different degrees of improvement in the prediction accuracy of each county. This improvement in western Liaoning was relatively more obvious. T&C&RS-M can accurately estimate the actual yield of different corn growing scenarios and have a high potential of yield estimation.

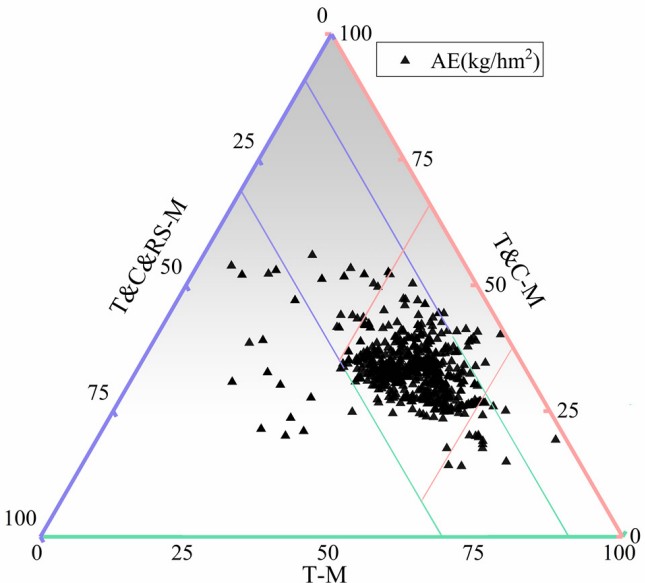

**Figure 6.** *AE* distribution of model estimation in counties from 2007 to 2016.

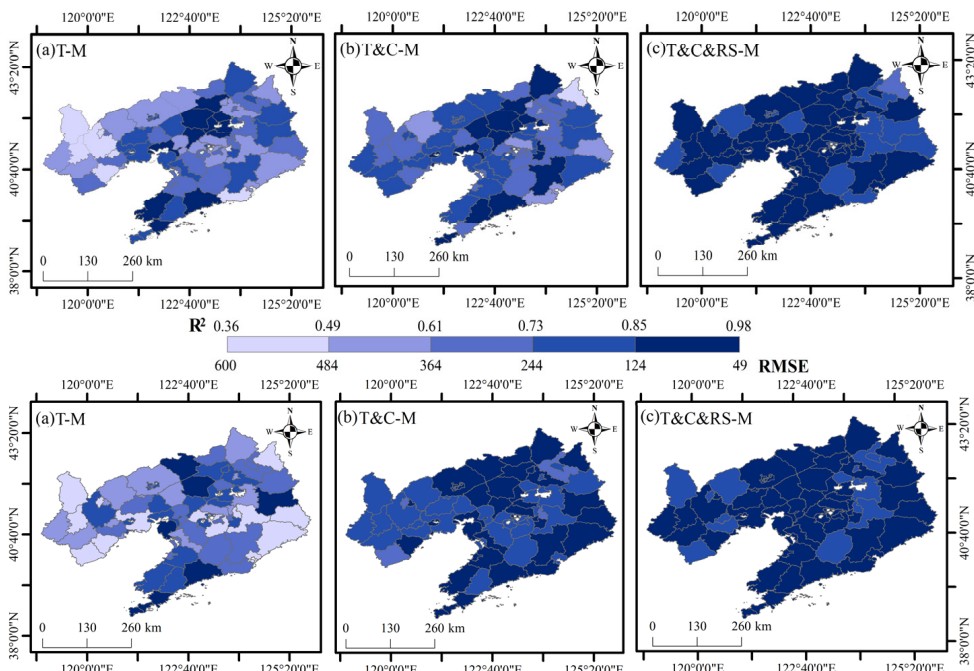

**Figure 7.** Comparison of the fitting effect of the model before and after coupling multi-source information ($R^2$ (**above**) and *RMSE* (**below**)).

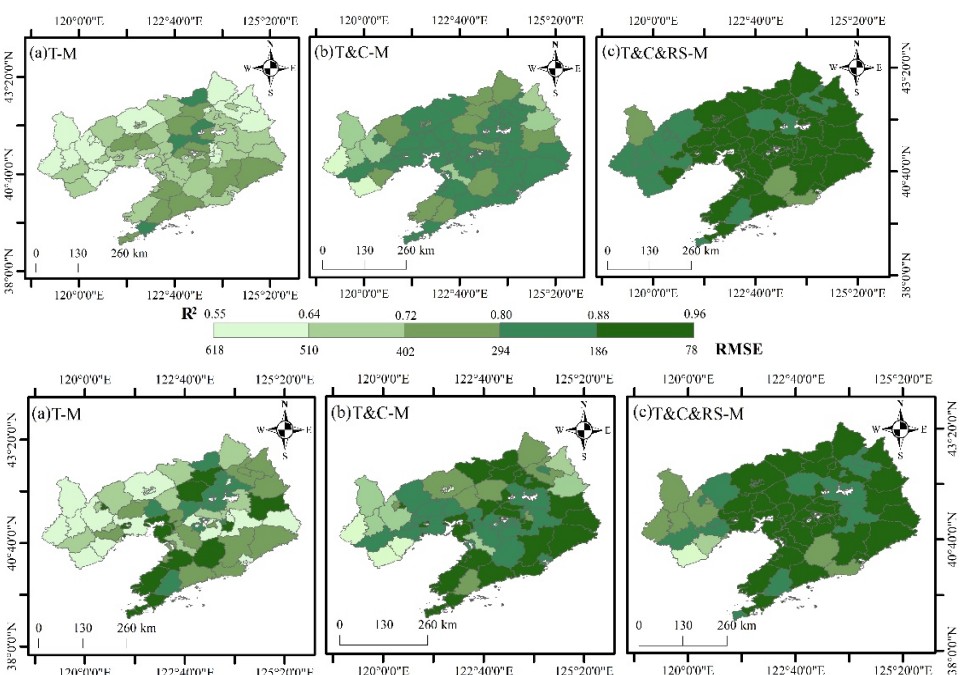

**Figure 8.** Comparison of the predictive application of the model before and after coupling multi-source information ($R^2$ (**above**) and *RMSE* (**below**)).

## 4. Discussion

The yield-estimation scheme in this study demonstrates its potential for accuracy and applicability due to the consideration of trend yield, related climate suitability, and vegetation index. T-M can capture trends in yield development but has less estimation accuracy. In counties where there are relatively stable changes in yield between adjacent years, such as Jinzhou City and Shenyang City, the predicted values of the three models do not differ significantly from the actual yields. This is due to the fact that artificial influences such as field management make the impact of climate change on yield minimal, and coupled with the absence of pests and diseases, T-M alone is sufficient to produce good results. However, for the condition with sudden decrease or increase in yield due to climate change or other factors, T-M suffers serious overestimation or underestimation phenomenon, such as counties in western Liaoning where drier phenomenon may occur, frequent abnormal weather conditions become the main cause of corn yield variation in these areas. Although T&C-M and T&C&RS-M also produce the phenomenon of overestimation and underestimation, they effectively improve this phenomenon. This is because T-M only considered elements such as field management and variety improvement, and failed to consider the effects of temperature, precipitation, sunshine, and other natural factors that vary in the short term on yield. In addition to considering the inherent effects of field management and crop varieties, T&C-M also accounts for yield fluctuations arising from climatic shifts, thereby offering a more comprehensive assessment of crop productivity than T-M. However, it does not consider the effects of other factors such as biotic and abiotic stressors, including pest infestations, pathogen infections, and environmental conditions which are difficult to capture by observatories, such as dry and hot winds. However, the impact of these factors on crop yields can be observed through changes in crop growth, which can be monitored using remote sensing means. Thus, the predicted outcomes of T&C-M for years affected by other factors, such as pests and agro-meteorological disaster, are less reliable than those of T&C&RS-M which combines multi-source data based on satellite, climate, and historical data. Remote sensing data provides valuable information on crop growth, making T&C&RS-M effective in various farming environments. For instance, in June 2017, during the critical corn jointing stage, corn fields of Huludao City and Kaiyuan City suffered damage from hailstorms, resulting in plant tearing and stunted growth. However,

T&C&RS-M was still able to make relatively accurate predictions, outperforming both T-M and T&C-M. The same was true of the insect attack at Kangping in 2012 and the hail damage at Jianping in 2014. Integrating multiple source indicators as inputs to the estimation model can provide valuable information on crop growth status and environmental stress, thereby improving the accuracy of yield estimation. This is consistent with other research findings [31,34,81,82]. We also performed the experiment with multiple counties in Liaoning to evaluate the applicability of the developed models in different regions. The robustness of our approach was confirmed through multi-year fitting, which gained similar performance results.

Large-scale crop yield estimation demands both accurate estimation results and feasible estimation methods. Although the crop growth mechanism models calibrated using ground measurement data have substantial advantages, the statistical model is still the estimation method with high application value in terms of current and future operability. Therefore, we will further strengthen the comparison of the model construction using different statistical methods and try to improve the model performance.

## 5. Conclusions

In this study, using the setting of the typical rain-fed model of Liaoning, we built a county-level corn yield estimation model coupled with multi-source data. In contrast to crop growing models with complicated operations, the model parameters in this study are simple to obtain, which can accommodate the requirements of rapid and sizeable regional scale yield estimation. The model construction process pays attention to the fit with the physiological characteristics of corn to enhance the mechanics of statistical models. The remote sensing information under macroscopic observation has the advantage of feedback on crop status. The introduction of this information has optimized the defects of the climate estimation model on the dynamic diagnosis of growth, effectively improving the accuracy. The $R^2$ reached 0.82 in predictive application validation, especially in the case of cold damage, hailstorm, pest, and other external environmental changes which may cause yield loss.

This study integrates multi-source information features and adds crop growth mechanisms to the statistical yield estimation model established based on mathematical relationships. However, the remote sensing data index system for evaluating crop growth still requires refinement. Regarding this issue, we should have many single-point sample experiments to extend from "point" to "plane". Researchers should seek scientific and quantitative evaluation of crop growth by remote sensing data and develop a closer integration between remote sensing technology and crop estimation.

**Author Contributions:** Conceptualization, G.Q. and Y.S.; methodology, G.Q. and Y.S.; data organization, G.Q. and X.P.; writing—original draft preparation, G.Q. and C.S.; writing—review and editing, J.H. and Y.S.; supervision, Y.S.; funding acquisition, Y.S. All authors have read and agreed to the published version of the manuscript.

**Funding:** This research was supported by the National Key Research and Development Program of China (no. 2020YFA0608501), the National Natural Science Foundation of China (no. 42071351), and the project-supporting discipline innovation team of Liaoning Technical University (no. LNTU20TD-23).

**Data Availability Statement:** Publicly available datasets were analyzed in this study. The data sources are indicated in the text.

**Acknowledgments:** We thank the support given to the authors by the Information Research Institute, Liaoning Academy of Agricultural Sciences. The corn yield data was provided by the Information Research Institute. We thank the relevant teams and organizations for providing the datasets used in this study. In addition, we thank the reviewers for their valuable suggestions that helped to improve the quality of the manuscript.

**Conflicts of Interest:** The authors declare no conflict of interest.

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
