# Peer review of "County Scale Corn Yield Estimation Based on Multi-Source Data in Liaoning Province"

_agronomy, doi:10.3390/agronomy13051428_

Round 1

Reviewer 1 Report

This is a very well english written manuscript outlining methods for estimate corn yield at county scale base don multi-data in Liaoning province of China. However, I have specific comments that should be addressed by the authors:

1)     in several parts of the introduction it says "Error! Reference source not found". Please check.

2)     In the methodological part, it is not clear why they apply a multiple regression analysis and the weight of factors to esgtimate the total yield.

3)     Instead of performing a multiple regression analysis by county, wouldn't it be convenient to aplly a mixed model using counties as random factors?

4)     It is not clear why they adjusted the model with the years 2007 to 2016 and used only the year 2017 to validate. I suggest running the model several times, always using a different set to fit the model and a different year to validate. In this way, it can be determined how sensitive is the fitted model.

5)     I suggest using the traditional structure of introduction-materials and methods-results and discussion.

6)     It is suggested to further expand the discussion of the results, highlighting which methodological aspects associated with remote sensing and climate data could improve the obtained results.

Author Response

The line numbers (at Line 168) in our responses are calibrated to the PDF version.

We are deeply thankful for the critiques and suggested experiments that have greatly improved our manuscript.

Review1:

This is a very well english written manuscript outlining methods for estimate corn yield at county scale base don multi-data in Liaoning province of China. However, I have specific comments that should be addressed by the authors:

Response: Thank you for raising these important points. We apologize for the confusion generated by the previous version of the manuscript and sincerely hope that our new version has been substantially improved. We also have now worked on both language and readability and have also involved native English speakers for language corrections.

1) in several parts of the introduction it says "Error! Reference source not found". Please check.

Response 1: Thanks for your reminder. We apologize for the error and it may be caused by incompatibility of document versions or system problem. We have carefully examined the new manuscript to avoid the error of reference source.

2) In the methodological part, it is not clear why they apply a multiple regression analysis and the weight of factors to esgtimate the total yield.

Response 2: Thanks, our methodological part was considered from the following perspectives.

Crop yields depend mainly on climate, soil, genotype and management factors. However, information pertaining to soil, genotype and management is difficult to monitor directly. In fact, the information can be derived indirectly by Vegetation Index (VI). Nonetheless, the applicability of VIs largely depends on crop type and local conditions. The VIs in different periods affect yield estimation to varying degrees. Only a few studies have systematically evaluated the sensitivity of VIs and climate factors to yield estimation but it helps to improve the accuracy (The details is given in Introduction).

Therefore, in this paper, we through importance evaluation of Random Forest, we determined the factor weights to obtain the different contributions of multiple variables when they jointly explain corn yield in Liaoning. The resulting weights were used as the original coefficients of the variables to cut the model into the estimation of corn yield in the Liaoning region, which initially achieved the goal of model localization. After determining the weight of factors targeting overall Liaoning, we respectively use Multiple Linear Regression (MLR) for each country to establish the linear relationship between the weighted factors and corresponding fluctuating yield, adding trend yield to complete final predicting process in building the county-level yield estimation model. This logical pattern that adjusted the model from overall situation to local is conducive to improving its adaptation.

We have revised the text to address your concerns and hope that it is now clearer. Please see at 2.3.4 and 3.1 section of the revised manuscript.

3) Instead of performing a multiple regression analysis by county, wouldn't it be convenient to apply a mixed model using counties as random factors?

Response 3: Thanks, performing a multiple regression analysis by county during model building was considered from the following perspectives.

Mixed models based on national or provincial scales can reflect changes in the macroscopic pattern of food production, but can hardly reveal the heterogeneity of yield changes within small regions. In Liaoning Province, topography, climate, soil properties and management measures are distinct in different counties (The details is given in 2.1. Study Area). Therefore, it is necessary to consider county characteristics when building yield estimation models. We take counties as the basic unit of study, from the evaluation of the importance of yield estimation factors for all counties in Liaoning to the construction of specific yield estimation equations for each county. We explored regional differences at the county scale from a holistic to a local perspective, avoiding the randomness of individual models.

On the other hand, yield estimation model in this paper consists of trend model, fluctuating yield model, and random error term (equation 1 and 10 in the manuscript). It is consistent with the construction concept of the mixed model Y= YA +YB +U0 (YA is factor A correlation model, YB is factor B correlation model, and U0 is error term). The model in this paper is simple and flexible in prediction, easy to implement, can meet the needs of building a wide and fast yield estimation model application.

4)  It is not clear why they adjusted the model with the years 2007 to 2016 and used only the year 2017 to validate. I suggest running the model several times, always using a different set to fit the model and a different year to validate. In this way, it can be determined how sensitive is the fitted model.

Response 4: We sincerely thank the Reviewer for their valuable feedback. We used the historical data back substitution confirmation to evaluate the fitting accuracy of the model (using the statistics yield from 2006-2016) and the extrapolation forecast test to evaluate the practical application accuracy of the model(using the statistics yield from 2017). We have added more specific descriptions at Line 508-525.

As suggested by the Reviewer, we have tried our best to use different years to validate model performance by the historical data back substitution confirmation and the extrapolation forecast test. We agree that more validation would be useful to understand the details of interaction and enhancement. At this point, if more county-level yield data with good quality can be obtained in the future, we will extend this exploration, and continue to accumulate and improve in practical applications.

5)  I suggest using the traditional structure of introduction-materials and methods-results and discussion.

Response 5: Thanks, we have modified the manuscript to the traditional structure of introduction-materials and methods-results and discussion following your suggestion.

6)    It is suggested to further expand the discussion of the results, highlighting which methodological aspects associated with remote sensing and climate data could improve the obtained results.

Response 6: Thanks, considering the Reviewer’s suggestion, we have supplemented the text and hope that it is now clearer. Please see page 18 of the revised manuscript, lines 574–613.

We would like to thank the referee again for taking the time to review our manuscript.

Reviewer 2 Report

This study addresses an important issue of timely and reliable corn yield estimation at a large scale, which is crucial for preventing climate risk and meeting the growing demand for corn. The use of publicly available data and a simple and scalable modeling approach is a significant contribution, as it can potentially be employed in situations with sparsely ground data for estimating crop yields. 

Some minor issues need to be addressed before publication

1. Please check the format thoroughly. Many references were missed.

2. Line 57. Liu et al., 2023 could be a good reference for APSIM simulation here. 

Liu, K., Harrison, M.T., Yan, H. et al. Silver lining to a climate crisis in multiple prospects for alleviating crop waterlogging under future climates. Nat Commun 14, 765 (2023). https://doi.org/10.1038/s41467-023-36129-4

3. Line 62. I think you were talking about machine learning model? crop growth models are process-based, not black box. 

4. Figure 4 caption needs to be clear e.g. is it CV or confidential interval?

5. Supply the full name of abbreviation in Fig. 7&8.

6. How about modelling performance in phenolog?

Author Response

The line numbers (at Line 168) in our responses are calibrated to the PDF version.

We are deeply thankful for the critiques and suggested experiments that have greatly improved our manuscript.

Please see attached file for point-to-point replies

Reviewer 3 Report

I have read the manuscript titled 'County scale corn yield estimation based multi-data in Liaoning Province', which describes and tests a method using both historical yield, climate and remote sensing data to estimate corn yield.

The introduction is well written (although there are many error messages for references), but the method description, section 3, lacks detail and is sometimes hard to understand. In particular, I find it often not clear what datasets are used. I have made a number of comments in the text that I will attach.

In general, the English language is understandable, but sections 3-5 in particular, would benefit from a language check.

In my opinion, the method presented in this manuscript is interesting and worth publishing. However, there is a need for further detail and explanation, and I recommend major revision.

Check the references, there are a lot of error messages, particularly in the introduction.

Check the references to equations, it looks like the numbers are wrong.

Author Response

The line numbers (at Line 168) in our responses are calibrated to the PDF version.

We are deeply thankful for the critiques and suggested experiments that have greatly improved our manuscript.

Please see attached file for point-to-point replies.

Round 2

Reviewer 1 Report

I found the new version adequate and the authors' response to all my observations was accurate and clear. For all these reasons, I consider that the manuscript is suitable for publication.